# LOTTERY AWARE SPARSITY HUNTING: ENABLING FEDERATED LEARNING ON RESOURCE-LIMITED EDGE

## ABSTRACT

Limited computation and communication capabilities of clients pose significant challenges in federated learning (FL) over resource-limited edge nodes. A potential solution to this problem is to deploy off-the-shelf sparse learning algorithms that train a binary sparse mask on each client with the expectation of training a consistent sparse server mask yielding sparse weight tensors. However, as we investigate in this paper, such naive deployments result in a significant drop in accuracy compared to FL with dense models, especially for clients with limited resource budgets. In particular, our investigations reveal a serious lack of consensus among the trained sparsity masks on clients, which prevents convergence for the server mask and potentially leads to a substantial drop in model performance. Based on such key observations, we propose *federated lottery aware sparsity hunting* (FLASH), a unified sparse learning framework to make the server win a lottery in terms of yielding a sparse sub-model, able to maintain classification performance under highly resource-limited client settings. Moreover, to support FL on different devices requiring different parameter density, we leverage our findings to present *hetero-FLASH*, where clients can have different target sparsity budgets based on their device resource limits. Experimental evaluations with multiple models on various datasets (both IID and non-IID) show superiority of our models in closing the gap with unpruned baseline while yielding up to $\sim 10.1\%$ improved accuracy with $\sim 10.26\times$ fewer communication costs, compared to existing alternatives, at similar hyperparameter settings. `Code is released as Supplementary.`

## 1 INTRODUCTION

Federated learning (FL) McMahan et al. (2017) is a popular form of distributed training, which has gained significant traction due to its ability to allow multiple clients to learn a shared global model without the requirement to transfer their private data. However, clients' heterogeneity and resource limitations pose significant challenges for FL deployment over edge nodes, including mobile phones and IoT devices. To resolve these issues, various methods have been proposed over the past few years including efficient learning for heterogeneous collaborative training Lin et al. (2020); Zhu et al. (2021), distillation He et al. (2020), federated dropout techniques Horvath et al. (2021); Caldas et al. (2018b), efficient aggregation for faster convergence and reduced communication Reddi et al. (2020); Li et al. (2020b). However, these methods do not necessarily address the growing concerns of highly computation and communication limited edge.

Meanwhile, reducing the memory, compute, and latency costs for deep neural networks (DNNs) in centralized training for their efficient edge deployment has also become an active area of research. In particular, recently proposed *sparse learning* (SL) strategies Evci et al. (2020); Kundu et al. (2021b); Mocanu et al. (2018); Dettmers & Zettlemoyer (2019); Raihan & Aamodt (2020) effectively train weights and associated binary *sparse masks* to allow only a fraction of model parameters to be updated during training, potentially enabling the lucrative reduction in both the training time and compute cost Qiu et al. (2021); Raihan & Aamodt (2020), while creating a *model to meet a target parameter density denoted as d, and is able to yield accuracy close to that of the unpruned baseline*.

However, the challenges and opportunities of sparse learning in FL is yet to be fully unveiled. Only very recently, few works Bibikar et al. (2021); Huang et al. (2022) have tried to leverage sparse learning in FL primarily to show their efficacy in non-IID settings. Nevertheless, these works primarily used sparsity for non-aggressive model compression,

limiting the actual benefits of sparse learning, and required multiple local epochs, that may further increase the training time for stragglers making the overall FL process inefficient Zhang et al. (2021). Moreover, the server-side pruning used in these methods may not necessarily adhere to the layers' pruning sensitivity[1] that often plays a crucial role in sparse model performance Kundu et al. (2021b); Zhang et al. (2018). Another recent work, ZeroFL Qiu et al. (2021), has explored deploying sparse learning in FL settings. However, Qiu et al. (2021) could not leverage any advantage of model sparsity in the clients'

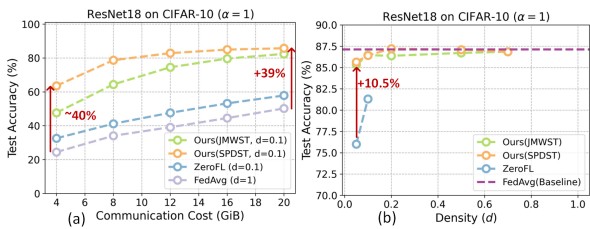

Figure 1: Comparison of (a) accuracy at different communication budget with, ZeroFL Qiu et al. (2021) and FedAvg. (w/ $d = 1.0$) (b) Accuracy vs. parameter density of each client. Proposed approaches can significantly outperform the existing alternative Qiu et al. (2021) at ultra-low target parameter density ($d$).

communication cost and had to keep significantly more parameters active compared to a target $d$ to yield good accuracy. Moreover, as shown in Fig. 1(b), for $d = 0.05$, ZeroFL still suffers from substantial accuracy drop of $\sim14\%$ compared to the baseline.

**Our Contributions.** Our contribution is fourfold. In view of the above limitations, we first identify crucial differences between a centralized and the corresponding FL model, in learning the sparse masks for each layer. In particular, we observe that in FL, the server model fails to yield convergent sparse masks, primarily due to the lack of consensus among clients' later layers' masks. In contrast, the centralized model show significantly higher convergence trend in learning sparse masks for all layers. We then experimentally demonstrate the utility of pruning sensitivity and mask convergence in achieving good accuracy setting the platform to close the performance gap in sparse FL.

We then leverage our findings and present *federated lottery aware sparsity hunting* (FLASH), a sparse FL methodology addressing the aforementioned limitations in a unified manner. At the core, FLASH leverages a two-stage FL, a robust and low-cost layer sensitivity evaluation stage and a FL training stage. In particular, the disentangling of the layer sensitivity evaluation from sparse weight training allows us to either choose to train a sparse mask or freeze a sensitivity driven pre-defined mask. This can further translate to a proportional communication saving.

To deal with the heterogeneity in clients' compute-budget, we further extend our methodologies to *hetero*-FLASH, where individual clients can support different density based on their resources. Here, to deal with the unique problem of the server selecting different sparse models for clients, we present server-side gradual mask sub-sampling, that identifies sparse masks via a form of layer sensitivity re-calibration, starting for models with highest to that with lowest density support.

We conduct experiments on MNIST, FEMNIST, and CIFAR-10 with different models for both IID and non-IID client data partitioning. Experimental results show that, compared to the existing alternative Qiu et al. (2021), at iso-hyperparameter settings, FLASH can yield up to $\sim8.9\%$ and $\sim10.1\%$, on IID and non-IID data settings, respectively, with reduced communication of up to $\sim10.2\times$.

## 2 RELATED WORKS

**Model Pruning.** Over the past few years, a plethora of research has been done to perform efficient model compression via pruning, particularly in centralized training Ma et al. (2021); Frankle & Carbin (2018); Liu et al. (2021); You et al. (2019); He et al. (2018). Pruning essentially identifies and removes the unimportant parameters to yield compute-efficient inference models. More recently, sparse learning Evci et al. (2020); Kundu et al. (2021b); Dettmers & Zettlemoyer (2019); Raihan & Aamodt (2020); Kundu et al. (2020; 2019), a popular form of model pruning, has gained significant traction as it can yield FLOPs advantage even during training. In particular, it ensures only $d\%$ of the model parameters remain non-zero during the training for a target parameter density $d$ ($d < 1.0$ and sparsity is $100 - d\%$), potentially enabling training compute and comm. cost if deployed for FL.

**Dynamic network rewiring (DNR).** We leverage DNR Kundu et al. (2021b), to sparsely learn the sparsity mask of each client. In DNR, a model starts with randomly initiated mask following the

---
[1] We measure layer importance via the proxy of sensitivity. A layer with higher sensitivity demands higher % of non-zero weights compared to a less sensitive layer.

target parameter density $d$. After an epoch, the client evenly prunes the lowest $p_r\%$ weights from each layer based on absolute magnitude, where $p_r$ is prune rate. Note, this $p_r\%$ pruning happens on top of the sparse model with density $d$, allowing $p_r\%$ weights to be regrown. DNR then ranks each layer based on the normalized contribution of the summed non-zero weight magnitudes. Finally, the client regrows total $p_r\%$ weights in a non-uniform way, allowing more regrowth to the layers having higher rank. This process iteratively repeats over epochs to finally learn the mask.

**Federated learning for resource and communication limited edge.** To address device heterogeneity, existing works have explored the idea of heterogeneous training Horvath et al. (2021); Diao et al. (2020); Yao et al. (2021) allowing different clients to train on different fractions of full-model based on their compute-budget. On a parallel track, various optimizations are proposed in FL training framework to accelerate convergence, thus requiring fewer communication rounds Han et al. (2020); Gorbunov et al. (2021); Zhang et al. (2013); Li et al. (2019); Reddi et al. (2020); Islamov et al. (2021); Albasyoni et al. (2020).

To address the issue of client resource limitations, a few research have leveraged pruning in FL Li et al. (2020a); Jiang et al. (2022); Li et al. (2021). In particular, LotteryFL Li et al. (2020a) trained each client to have their personalized mask with which they are able to perform well only on their own data. Moreover, the clients often need to send full model costing bandwidth. PruneFL Jiang et al. (2022) also asks for significant communication costs as it demands participating clients to send all the gradient values to the server while updating the masks.

Only a few contemporary works Huang et al. (2022); Bibikar et al. (2021); Qiu et al. (2021) tried to leverage the benefits of sparse learning in federated settings. In particular, Huang et al. (2022) relied on a randomly initialized sparse mask, and recommended keeping it frozen throughout the training, yet failed to provide any supporting intuition. FedDST Bibikar et al. (2021), on the other hand, leveraged the idea of RigL Evci et al. (2020) to perform sparse learning of the clients, relied on a large number of local epochs to avoid gradient noise, and focused primarily on only highly non-IID data without targeting ultra-low density $d$. More importantly, neither of these works investigated the key differences between centralized and FL sparse learning. With similar philosophy as ours, ZeroFL Qiu et al. (2021) first identified a key aspect of sparse learning in FL in terms of all clients' masks to be within $30\%$ of the total model weights to yield good accuracy at high compression. However, ZeroFL suffered significantly in exploiting a proportional advantage in communication saving as even for low parameter density $d$, all clients had to download the dense model and send back a $3\times$ denser model. Furthermore, these algorithms sacrifice significant accuracy at ultra-low $d$.

## 3 REVISITING SPARSE LEARNING: WHY DOES IT MISS THE MARK IN FL?

Table 1: FL training settings considered in this work.

| Dataset | Model | #Params. | Data-partioning | Rounds $(T)$ | Clients $(C_N)$ | Clients/Round $(c_r, c_d)$ | Optimizer | Aggregation type | Local epoch $(E)$ | Batch size |
|---------|-------|----------|-----------------|--------------|-----------------|----------------------------|-----------|------------------|-------------------|------------|
| MNIST | MNISTNet | 262K | LDA | 400 | 100 | 10, 10 | | | | 32 |
| CIFAR-10 | ResNet18 | 11.2M | | 600 | | | SGD | FedAvg | 1 | 32 |
| FEMNIST | Same as Caldas et al. (2018a) | 6.6M | Reddi et al. (2020) | 1000 | 3400 | 34, 34 | | McMahan et al. (2017) | | 16 |

Note, centralized training has shown significant benefits with sparse learning in FLOPs reduction during forward operations Evci et al. (2020), and potential training speed-up of up to $3.3\times$ Qiu et al. (2021) while maintaining close to the baseline accuracy, even at $d \leq 0.1$. We now use a sparse learning, namely Kundu et al. (2021b), in FL settings (refer to Table 1 for details) on CIFAR-10, where each client separately performs Kundu et al. (2021b) to train a sparse server-side ResNet18 and meet a fixed parameter density $d$, starting from a random sparse mask. After sending the updates to server, it aggregates them using FedAvg. We term this as *naive sparse training* (NST). Note, due to lack of knowledge about the pruning sensitivity for each layer, the server fails to sub-sample from the aggregated weights to meet target non-zero parameter density $d$. Thus, the down-link communication cost is higher as generally the aggregated non-zero parameter density is $> d$.

**Observation 1.** *At high compression $d \leq 0.1$, the collaboratively learned FL model significantly sacrifices performance, while the centralized sparse learning yields close to baseline performance.*

As shown in Fig.2(a), naive deployment of sparse learning significantly sacrifices accuracy in FL. In particular, for $d = 0.1$, the trained server-side model suffers an accuracy drop of $3.67\%$. At even lower $d = 0.05$, this drop significantly increases to $12.03\%$, hinting at serious limitations of sparse

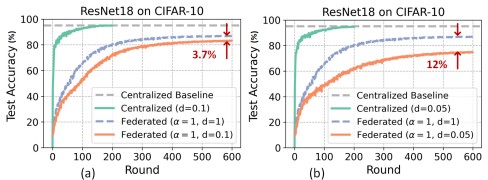 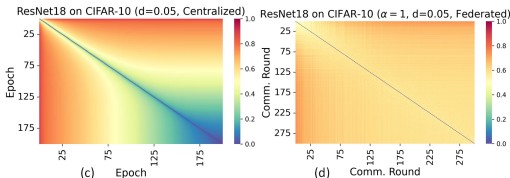

Figure 2: (a-b) Accuracy vs. round plot on deployment of off-the-shelf sparse learning in FL for different $d$, (c-d) visualization of the Model's SM in terms of Jaccard distance while training with sparse learning for (c) centralized and (d) FL, respectively.

learning in FL. However, an off-the-shelf centralized sparse learning can yield model having close to the baseline accuracy, even at $d = 0.05$.

**Observation 2.** *As the training progresses, the sparse masks in centralized training tend to agree across epochs, showing convergence, while the server mask in FL does lack agreement across rounds.*

**Definition 1. Sparse mask mismatch.** For a model at round $t$, we define the *sparse mask mismatch* (SM) $\mathtt{sm}^t$ as the Jaccard distance that is measured as follows.

$$\mathtt{sm}^t = 1 - \frac{(\sum_{l=1}^{L} \mathcal{M}_l^t \cap \mathcal{M}_l^{t-1})}{(\sum_{l=1}^{L} \mathcal{M}_l^t \cup \mathcal{M}_l^{t-1})} \tag{1}$$

where $\mathcal{M}_l^t$ represents the sparse mask tensor for layer $l$ at the end of round $t$. Interestingly, as depicted in Fig. 2(a), the SM for centralized learning tends to zero as the training progresses. In contrast, with the same model, dataset and $d$ values, in FL, the SM remains $> 0.4$ indicating a substantial distinction in the sparse mask learning between centralized and federated learning.

**Observation 3.** *At low target density, in federated sparse learning, throughout the training rounds, the disagreement on the later layer's masks remains more severe than the earlier ones.*

As the training progresses, in centralized learning, mask for each layer shows significant convergence trend as measured by SM for the layer (Fig. 3(a)). However, Fig. 3(b) shows in FL, the later layers' masks differ significantly and continue to disagree over rounds with SM value as high as $\sim$0.8. This may be attributed to many possible mask choices due to the later layers' significantly fewer non-zero parameter allocation, compared to the initial layers, driven by their respective pruning sensitivities. For example, layer 1 requires 90% parameters to be present, compared to only 5% for layer 14, with the later costing an SM of $\sim$0.73.

Table 2: Performance based on the different levels of mask disagreement in centralized.

| Training type | Use sensitivity | Masks change at | Layer SM | Test acc% |
|---|---|---|---|---|
| Pre-defined w/ mask frozen | N | – | – | 89.72 |
| Pre-defined w/ mask frozen | Y | – | – | **91.66** |
| | Y | layer 9-16 | 0.8 | 88.88 |
| w/o mask frozen | Y | layer 1-16 | 0.5 | 84.62 |
| | Y | layer 1-16 | 0.8 | 82.32 |

To further investigate the impact of higher SM and layer sensitivity on a model's accuracy, we performed five different training in centralized as described in Table 2. In particular, for the training in row1 we randomly generate sparse masks with uniform density for all the layers. For all other training, we first randomly create each layer's mask by following its pruning sensitivity[2] and then decide to keep the layer mask frozen for some or all the layers. For training described in rows 3-5, we allow a fraction of the mentioned layers' masks to differ between consecutive epochs such that they meet the target SM value, creating the situation of non-convergent masks. As Table 2 clearly shows that large SM for the layers can degrade the accuracy by up to $9.34\%$, we can safely conclude that *disagreement of masks across epochs can significantly affect the model's final performance*. Moreover, the model trained via sparse learning with sensitivity-driven pre-defined masks yields better performance than the one trained with uniform density sparse mask.

## 4 FLASH: METHODOLOGY

To win a lottery of having a sparse network yielding high accuracy at reduced parameters, we identify two key characteristics of sparse learning, namely, the pruning sensitivity and mask learnability towards convergence. To explicitly adhere to these two important aspects, in FLASH, we present a

---

[2]For a sparse model it is evaluated as the ratio $\frac{\texttt{\# of non-zero layer parameters}}{\texttt{\# layer parameters}}$ Ding et al. (2019). We use another pre-trained model of the same architecture and target $d$ for this evaluation.

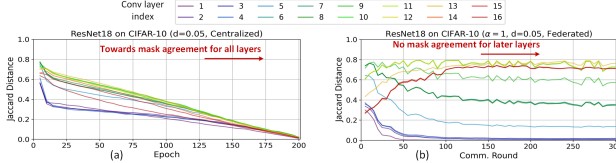

Figure 3: Layer-wise sparse mask mismatch (SM) vs. training epochs (rounds) plot for (a) centralized and (b) FL, respectively. In FL, the layer layers continue to have higher SM contrary to centralized, where every layer tend to reduce the SM as the training matures.

two-stage sparse FL method, stage 1: targeting sensitivity analysis to identify good initial sparse mask for each layer, stage 2: targeting training to learn masks and weights. In particular, to evaluate layer sensitivity in stage 1, the server randomly selects a small fraction of clients ($[\mathcal{C}_d]$), each locally sparse learning Kundu et al. (2021b) for few warm-up epochs ($E_d$) (L4-9 in Algo. 1). Upon collection of layer-wise sensitivity from the clients, for each layer $l$, the server estimates average density[3] $\hat{d}^l$ as $\frac{\sum_{i=1}^{c_d} d_i^l}{c_d}$, where $d_i^l$ is the density at layer $l$ in $i^{th}$ client. As these averaged layer-wise density values may not necessarily yield to the target density $d$, for a model with $K$ parameters we follow the following *density re-calibration*

$$d_c^l = \hat{d}^l.r_f, \text{ where } r_f = \frac{d \times K}{\sum_{l=1}^{L} \hat{d}^l.k^l} \tag{2}$$

$k^l$ is dense model's parameter size for layer $l$. For each layer $l$ of the model, the server then creates a binary sparse mask tensor that is randomly initialized, with a fraction of 1s $\propto d_c^l$ (L10). In stage 2, the server begins the training rounds starting with a sparse model initialization following the sparse mask computed in stage 1 (L11). Particularly, at each round, the clients perform sparse learning for $E$ epochs *(L23-29)* with the choice to either train the mask or keep training on the weights with the mask frozen (L26). The later allows the clients to intermittently share masks saving up-link cost.

However, in FL settings, the masks often show poor convergence (section 3, Obs. 2). To address this, in stage 2, we present two sparse FL methods depending on the mask learnability being disabled or enabled (L13). The disabled scenario ($m_{freez} = 1$) essentially translates to sparse learning with pre-defined layer masks at initialization, allowing to only learn the weights and forcing all the clients to use the same initialized masks. This guarantees no mask divergence issue ($sm^t = 0$ for all $t$). Moreover, as FLASH disentangles the sensitivity evaluation stage from the training, the pre-defined mask in this scenario benefits from the notion of layer sensitivity. We thus aptly name this scenario *sensitivity-driven pre-defined sparse training* (SPDST). Interestingly, earlier research Bibikar et al. (2021) hinted at poor model performance with pre-defined masks, contrasting ours where we see significantly improved model performance, implying the importance of stage 1 (as will be elaborated in section 5).

In the enabled mask learning scenario ($m_{freez} = 0$), model masks and weights are jointly learned during clients' local learning, thus termed as *joint mask weight sparse training* (JMWST). However, as highlighted earlier, clients' naive sparse mask selection at the beginning of each round costs a considerable accuracy drop (section 3 Obs. 1). JMWST allows the server to select a sparse model for the clients at round $t + 1$. For clients' target density $d$, the aggregated server model *(L20)* at the end of round $t$, generally has density $d_S > d$. To enable efficient sampling of sparse model, we leverage the density re-calibration strategy (Eq. 2) by taking the $t^{th}$ round's clients' sensitivity into consideration. We then perform magnitude pruning to retain the top-$d_c^l$ fraction of parameters for $l^{th}$ layer at the server and send the pruned model to clients at round $t$ *(L21)*. Intuitively, such sampling of non-zero weights by the server reduces chances of wasted updates, and allows the layer masks to converge faster due to alignment with the layers' pruning sensitivity. The clients then perform local sparse learning, yielding another set of sparse models and so on. Note, the aggregation and sampling is simpler in SPDST, as the server model always remains at density $d$. In terms of yielding convergent masks, we indeed observed a lower SM for JMWST by ∼85% compared to that in NST, evaluated after 300 rounds on CIFAR-10. Algorithm 1 details the FLASH training methods. It is noteworthy that, the clients are only allowed to update mask after an interval of $r_{int}$, rounds, which for JMWST is set to 1 by default, allowing the server to evaluate masks at the end of every round.

**Extension to support heterogeneous parameter density.** To support different density budgets for different clients, we now present hetero-FLASH. Let us assume a total of $N$ support densities

---

[3]which is same as sensitivity for a layer.

---

**Algorithm 1:** FLASH Training.

**Data:** Training rounds $T$, local epochs $E$, client set $[\mathcal{C}_N]$, clients per rounds $c_r$, target density $d$, sensitivity warm-up epochs $E_d$, density warm up client count $c_d$, initial value of freeze masks $m_{freez} = 0$, training algorithm $A$ and Aggregation type $Agr$.

1   $\mathcal{M}^{init} \leftarrow$ createRandomMask$(d)$
2   $\Theta^{init} \leftarrow$ initMaskedWeight$(\mathcal{M}^{init})$
3   serverExecute:
4   # Calculate the layer-wise sensitivity in stage 1
5   Randomly sample $c_d$ clients $[\mathcal{C}_d] \subset [\mathcal{C}_N]$
6   **for** *each client $c \in [\mathcal{C}_d]$ in parallel* **do**
7     |   $\Theta_c \leftarrow$ clientExecute$(\Theta^{init}, E_d, 0)$ # $m\_freeze = 0$
8     |   $\mathcal{S}_c \leftarrow$ computeSensitivity$(\Theta_c)$
9   **end**
10 # Initialize a sensitivity-driven mask
11 $\mathcal{M}^0 \leftarrow$ initMask$([\mathcal{S}_c], d)$
12 $\Theta^0 \leftarrow$ initMaskedWeight$(\mathcal{M}^0)$
13 $m_{freez} \leftarrow$ freezeMask$(A)$#set to 1, and 0 for SPDST, and JMWST, respectively
14 # Start Stage 2
15 **for** *each round $t \leftarrow 1$* **to $T$ do**
16     Randomly sample $c_r$ clients $[\mathcal{C}_r] \subset [\mathcal{C}_N]$
17     **for** *each client $c \in [\mathcal{C}_r]$ in parallel* **do**
18       |   $\Theta_c^t \leftarrow$ clientExecute$(\Theta^{t-1}, E, m_{freez})$
19     **end**
20     $\Theta_S^t \leftarrow$ aggrParamUpdateMask$([\Theta_c^t], Agr)$
21     $\Theta^t \leftarrow$ subsampleServerModel$(\Theta_S^t, [\Theta_c^t], d, m_{freez})$
22 **end**
23 clientExecute$(\Theta_c, E, m_{freez})$:
24 $\Theta_{c0} \leftarrow \Theta_c$
25 **for** *local epoch $i \leftarrow 1$* **to $E$ do**
26     |   $\Theta_{ci} \leftarrow$ doSparseLearning$(\Theta_{ci-1}, m_{freez})$
27     |   $m_{freez} \leftarrow$ checkUpdateMask$()$
28 **end**
29 return $\Theta_{cE}$

---

$d_{set} = [d_1, .., d_M]$, where $d_i < d_{i+1}$. Now, for hetero-SPDST, we perform a sensitivity warm-up, to create the masks for the clients' with the highest density $d_N$. For any other density $d_i$, we sample a sparse mask from that with density $d_{i+1}$. Note, while creating the mask from $d_{i+1}$ to $d_i$, we follow the layer-wise density re-calibration approach as mentioned earlier. For hetero-JMWST, at the beginning of each round, the server performs magnitude pruning to yield $N$ sub models meeting $N$ different density levels, contrasting to the creation of one model in JMWST. Participating clients of different densities use the corresponding sub models to start their local sparse training. In hetero-FLASH, server performs aggregation by following a form of *weighted fed averaging* (WFA). In particular, with similar inspiration as Diao et al. (2020), to give equal importance to each parameter update in such heterogeneous settings, WFA averages the values by their number of non-zero occurrences among the participating clients. We have provided the algorithm for hetero-FLASH in the Appendix.

## 5   EXPERIMENTS

**Datasets and Models.** We evaluated the performance of FLASH on MNISTLeCun & Cortes (2010), Federated EMNIST (FEMNIST) Caldas et al. (2018a), and CIFAR-10 Krizhevsky et al. (2009) datasets with the CNN models described in McMahan et al. (2017), Caldas et al. (2018a), and ResNet18, respectively. Further model details are provided in the Appendix. For data partitioning of MNIST and CIFAR-10, we use Latent Dirichlet Allocation (LDA)Reddi et al. (2020) with three different $\alpha$ ($\alpha = 1000$ for IID and $\alpha = 1$ and $0.1$ for non-IID). For FEMNIST, we employ the same setting as in Han et al. (2020), which partitions the data based on the writer into 3400 clients, making it inherently non-IID.

**Training Hyperparameters.** We use Clients' starting learning rate ($\eta_{init}$) as 0.1 that is exponentially decayed to 0.001 ($\eta_{end}$) at the end of training. Specifically, learning rate for participants at round t is $\eta_t = \eta_{init}(\exp(\frac{t}{T}\log(\frac{\eta_{init}}{\eta_{end}})))$. In all the sparse learning experiments, prune rate is set to $0.25$[4]. Summary of the rest of the training hyperparameters can be found in 1. Furthermore, all the experiments were performed with three different seeds. We report the final results as the averaged accuracy with corresponding std deviation in the tables.

---

[4]Prune rate controls the fraction of non-zero weights participating in the redistribution during sparse learning.

Table 3: Results with FLASH (SPDST, and JMWST) and its comparison with NST and PDST.

| Dataset | Data Distribution | Density (d) | Baseline Acc % | NST Acc % | PDST Acc % | SPDST Acc % | JMWST($r_{int}=1$) Acc % | JMWST($r_{int}=5$) Acc % |
|---|---|---|---|---|---|---|---|---|
| MNIST | IID ($\alpha=1000$) | 1.0 | $98.79 \pm 0.06$ | – | – | – | – | – |
| | | 0.1 | – | $97.57 \pm 0.11$ | $97.09 \pm 0.18$ | $\mathbf{98.21 \pm 0.06}$ | $97.95 \pm 0.16$ | $98.09 \pm 0.16$ |
| | | 0.05 | – | $95.19 \pm 0.56$ | $94.8 \pm 1.04$ | $\mathbf{97.46 \pm 0.14}$ | $97.24 \pm 0.21$ | $97.37 \pm 0.23$ |
| | non-IID ($\alpha=1.0$) | 1.0 | $98.76 \pm 0.06$ | – | – | – | – | – |
| | | 0.1 | – | $97.36 \pm 0.19$ | $96.82 \pm 0.25$ | $97.96 \pm 0.13$ | $97.72 \pm 0.12$ | $\mathbf{98.11 \pm 0.12}$ |
| | | 0.05 | – | $95.75 \pm 0.31$ | $95.34 \pm 0.77$ | $97.3 \pm 0.26$ | $97.38 \pm 0.11$ | $\mathbf{97.59 \pm 0.07}$ |
| | non-IID ($\alpha=0.1$) | 1.0 | $98.45 \pm 0.17$ | – | – | – | – | – |
| | | 0.1 | – | $96.19 \pm 0.22$ | $94.41 \pm 1.23$ | $\mathbf{97.22 \pm 0.43}$ | $96.53 \pm 0.19$ | $96.7 \pm 0.14$ |
| | | 0.05 | – | $91.66 \pm 1.74$ | $91.06 \pm 1.1$ | $95.7 \pm 0.37$ | $95.83 \pm 0.84$ | $\mathbf{95.91 \pm 0.64}$ |
| CIFAR-10 | IID ($\alpha=1000$) | 1.0 | $88.56 \pm 0.06$ | – | – | – | – | – |
| | | 0.1 | – | $84.89 \pm 0.26$ | $86.72 \pm 0.09$ | $\mathbf{88 \pm 0.28}$ | $87.62 \pm 0.35$ | $87.86 \pm 0.13$ |
| | | 0.05 | – | $77.48 \pm 0.54$ | $84.38 \pm 0.12$ | $86.99 \pm 0.14$ | $86.87 \pm 0.08$ | $\mathbf{87.18 \pm 0.09}$ |
| | non-IID ($\alpha=1.0$) | 1.0 | $87.13 \pm 0.18$ | – | – | – | – | – |
| | | 0.1 | – | $83.46 \pm 0.19$ | $85.07 \pm 0.24$ | $\mathbf{86.42 \pm 0.49}$ | $\mathbf{86.45 \pm 0.31}$ | $86.36 \pm 0.13$ |
| | | 0.05 | – | $75.1 \pm 0.76$ | $83.33 \pm 0.14$ | $85.64 \pm 0.58$ | $85.34 \pm 0.27$ | $\mathbf{85.9 \pm 0.24}$ |
| | non-IID ($\alpha=0.1$) | 1.0 | $77.64 \pm 0.49$ | – | – | – | – | – |
| | | 0.1 | – | $71.18 \pm 1.23$ | $74.82 \pm 0.72$ | $\mathbf{76.74 \pm 1.46}$ | $74.74 \pm 1.07$ | $75.47 \pm 1.18$ |
| | | 0.05 | – | $61.29 \pm 2.76$ | $72.32 \pm 1.05$ | $75.47 \pm 2.31$ | $73.9 \pm 1.45$ | $\mathbf{75.49 \pm 0.9}$ |
| FEMNIST | non-IID | 1.0 | $84.68 \pm 0.20$ | – | – | – | – | – |
| | | 0.1 | – | $76.92 \pm 0.42$ | $76.01 \pm 1.26$ | $82.70 \pm 0.26$ | $83.02 \pm 0.21$ | $\mathbf{83.4 \pm 0.26}$ |
| | | 0.05 | – | $61.9 \pm 2.6$ | $63.65 \pm 0.86$ | $81.18 \pm 0.36$ | $82.01 \pm 0.53$ | $\mathbf{82.48 \pm 0.18}$ |

Table 4: Comparison of ZeroFL on various performance metrics with existing alternative sparse federated learning schemes. The italicized values are taken from the original manuscript.

| Dataset | Data Distribution | Method | Density | Acc% | Down-link Savings | Up-link Savings |
|---|---|---|---|---|---|---|
| CIFAR-10 | IID | ZeroFL Qiu et al. (2021) | 0.1 | $82.71 \pm 0.37$ | $1\times$ | $1.6\times$ |
| | | FLASH-SPDST (ours) | 0.1 | $\mathbf{88 \pm 0.28}$ | $\mathbf{9.8\times}$ | $\mathbf{9.8\times}$ |
| | | ZeroFL Qiu et al. (2021) | 0.05 | $78.22 \pm 0.35$ | $1\times$ | $1.9\times$ |
| | | FLASH-SPDST (ours) | 0.05 | $\mathbf{86.99 \pm 0.14}$ | $\mathbf{19.5\times}$ | $\mathbf{19.5\times}$ |
| | non-IID ($\alpha=1.0$) | ZeroFL Qiu et al. (2021) | 0.1 | $81.04 \pm 0.28$ | $1\times$ | $1.6\times$ |
| | | FLASH-SPDST (ours) | 0.1 | $\mathbf{86.42 \pm 0.49}$ | $\mathbf{9.8\times}$ | $\mathbf{9.8\times}$ |
| | | ZeroFL Qiu et al. (2021) | 0.05 | $75.54 \pm 1.15$ | $1\times$ | $1.9\times$ |
| | | FLASH-SPDST (ours) | 0.05 | $\mathbf{85.64 \pm 0.58}$ | $\mathbf{19.5\times}$ | $\mathbf{19.5\times}$ |
| FEMNIST | non-IID | ZeroFL Qiu et al. (2021) | 0.05 | $77.16 \pm 2.07$ | $1\times$ | $\mathbf{17.7\times}$ |
| | | FLASH-SPDST (ours) | 0.05 | $\mathbf{81.18 \pm 0.36}$ | $\mathbf{14.6\times}$ | $14.6\times$ |

## 5.1 EXPERIMENTAL RESULTS WITH FLASH

To understand the importance of `stage 1` in FLASH methodology, we identify a baseline training with uniform layer sensitivity driven *pre-defined sparse training* (PDST) in FL. Table 3 details the performance of FLASH at different levels of $d$, for various choices of sparse learning methods. In particular, as we can see in Table 3 column 5 and 6, the performance of both NST and PDST produced models cost heavy accuracy drop at ultra low parameter density $d = 0.05$. For example, on CIFAR-10 ($\alpha = 0.1$), models from NST and PDST sacrifice an accuracy of $16.35\%$ and $5.32\%$, respectively. However, at comparatively higher density ($d = 0.1$), both can yield models with a lower accuracy difference from the baseline by around $6.46\%$ and $2.82\%$. SPDST, on the other hand, can maintain **close to the baseline accuracy** at even ultra-low density for all data partitions. Interestingly, for majority of the cases, it even outperforms JMWST yielded models. *These results clearly highlight the efficacy of both sensitivity driven sparse learning (as SPDST > PDST) and early mask convergence (as SPDST ≈ JMWST) in FL settings. Importantly, for increased $r_{int}$ in JMWST, we observe a consistent improvement in accuracy.* The inferior accuracy at $r_{int} = 1$ can be attributed to the mask divergence caused by frequent noisy gradient dependent update. We thus believe efficient hyperparameter search including $r_{int}$ is essential for sparse FL model's improved performance, particularly for JMWST. Moreover, JMWST requires additional communication of non-zero weight indices, contrasting SPDST, where clients do not need to send the mask at all, *allowing us to yield proportional communication saving as the model density*. Fig. 4 shows the acc. vs. round comparison among the two proposed methods on different data distributions.

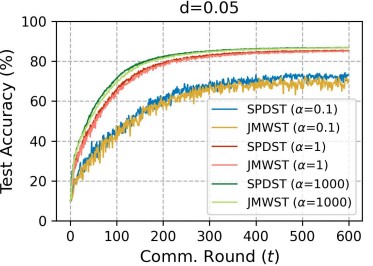

Figure 4: Test accuracy vs. round for different approaches on CIFAR-10.

**Comparison with ZeroFL.** Despite leveraging a form of sparse learning Raihan & Aamodt (2020), ZeroFL required significantly higher up-link/down-link communication cost compared to the target density $d$. This enables FLASH to gain a significant advantage in communication saving over ZeroFL, particularly for SPDST, as it only asks for the reduced size parameters to be communicated between the server and clients. In particular, we evaluate the communication saving as the ratio of the dense model size and corresponding sparse model size with the tensors represented in compressed sparse

Table 5: Performance of hetero-FLASH on various datasets where each client can have a density from the set $d_{set} \in [0.1, 0.15, 0.2]$.

| Dataset | Data Distribution | Max Client Density $(d_{set})$ | Hetero-SPDST Acc % | Hetero-JMWST $(r_{int} = 1)$ Acc % | Hetero-JMWST $(r_{int} = 5)$ Acc % |
|---------|------------------|-------------------|---------------------|-----------------|-----------------|
| MNIST | IID ($\alpha = 1000$) | 0.2 | **98.29 ± 0.05** | 97.44 ± 0.23 | 97.83 ± 0.10 |
| | non-IID ($\alpha = 1.0$) | | **98.29 ± 0.09** | 97.47 ± 0.22 | 97.80 ± 0.23 |
| | non-IID ($\alpha = 0.1$) | | **97.63 ± 0.22** | 96.11 ± 0.75 | 96.25 ± 0.86 |
| CIFAR-10 | IID ($\alpha = 1000$) | 0.2 | 87.19 ± 0.26 | 86.37 ± 0.2 | **87.39 ± 0.15** |
| | non-IID ($\alpha = 1.0$) | | 86.16 ± 0.04 | 84.67 ± 0.06 | **86.19 ± 0.24** |
| | non-IID ($\alpha = 0.1$) | | **75.23 ± 1.26** | 71.3 ± 2.75 | 74.34 ± 0.85 |
| FEMNIST | non-IID | 0.2 | **82.58 ± 0.24** | 82.2 ± 0.42 | 82.5 ± 0.55 |

row (CSR) format Tinney & Walker (1967). As depicted in Table 4[5], FLASH can yield an accuracy improvement of up to $10.1\%$ at a reduced communication cost of up to $10.26\times$ (computed at up-link when both send sparse models).

## 5.2 EXPERIMENTAL RESULTS WITH HETERO-FLASH

Table 5 shows the performance of hetero-FLASH where the clients can have three possible density budgets as defined by the $d_{set}$. We assume the maximum capacity clients' density budget of $0.2$. To train on all the density values, we first create three sets, each having $40\%$, $30\%$, and $30\%$ of total clients, and corresponds to density $0.2$, $0.15$, and $0.1$, respectively. Now, during every round, we sample $10\%$ from each set with corresponding target density. Similar to the trend in FLASH, hetero-SPDST outperforms the JMWST counter-parts by up to $3.93\%$ evaluated on the three datasets. Also, following similar trend as with homogeneous density clients, with increased mask update interval ($r_{int}$), the performance of hetero-JMWST gets a significant boost in accuracy of up to $3.04\%$.

## 5.3 QUANTITATIVE ANALYSIS

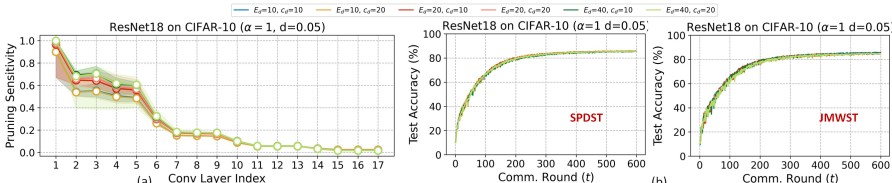

Figure 5: (a) Layer sensitivity evaluated at the end of sensitivity warm-up stage for different client participation size and their local epochs, (b) Comparison of server side model performance with the initialized sparse mask based on different sensitivity evaluated from (a).

**Dependence of initial sensitivity warm-up of participating clients.** To understand the importance of the clients' participation in the warm-up, we experimented with six different scenarios. In particular, we used two different values of participating clients ([10, 20]) each corresponding to three different local epoch choices ([10, 20, 40]). As shown in Fig. 5(a), the yielded pruning sensitivity follows a similar trend. Moreover, an SPDST training with mask chosen from any of these sensitivity lists finally yield FL models with similar performances (Fig. 5(b)), clearly demonstrating the robustness of our warm-up based sensitivity evaluation `stage 1`.

**Comparison with ERK+ initialization.** We now compare our SPDST mask initialization, with that of parameter density distribution evaluated via ERK+ Huang et al. (2022); Evci et al. (2020). Notably, contrary to uniform density, ERK+ scheme keeps more weights for the layers having fewer parameters. Note here, we use SPDST, ERK+, or uniform (PDST) as the initial mask for `stage 2`, and keep the mask frozen throughout the training of `stage 2`. As shown in Fig. 6(a-b), the *mask evaluation `stage 1` to initialize mask allows SPDST to consistently provide superior results over the other two*. We hypothesize this to the better layer sensitivity evaluation scheme of SPDST, particularly at the earlier layers, allowing it to retain more information at these layers.

**Importance of parameters' weighted fed averaging at the server.** Earlier literature Diao et al. (2020) suggested a form of weighted fed averaging, for clients with different model sizes. Inspired by that, we now investigate the necessity of WFA in FLASH. In particular, we performed experiments

---

[5]We understand for FEMNIST, ZeroFL reported significantly higher up-link saving, however, to the best of our understanding it should be similar to their report on other datasets, i.e. $\sim 1.9\times$.

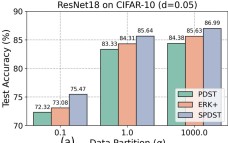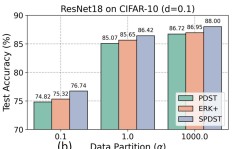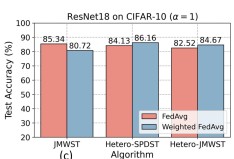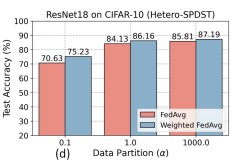

Figure 6: (a)-(b) Performance comparison of sparse models trained with SPDST, uniform (PDST) and ERK+ initialized layer-wise parameter density; (c-d)Performance comparison of fedavg with weighted fedavg for (c) different training algorithms and (d) different dataset partitioning ($\alpha$).

on CIFAR-10 ($\alpha = 1.0$), both with and without WFA, during server aggregation. As shown in Fig. 6(c), *WFA model performs inferior to the fed averaged model in FLASH. On the contrary, hetero-FLASH, enjoys consistent superior performance with WFA* 6(c-d). The inferior performance of WFA in FLASH may hint at the fact that if a weight is non-zero only for fewer clients, as compared to other non-zero weights, giving it equal weight as the others nullifies its lower importance, that may be necessary to preserve for mask convergence. On the other hand, having WFA in hetero FLASH is necessary, as a weight's less frequent non-zero occurrence can be due to fewer number of high-parameter density clients in a round. Further investigations on the utility and use case of such weighted averaging is an interesting future research direction.

**Time and communication overhead for `stage 1`.** Mask evaluation `stage 1` uses one round with $E_d$ local epochs (for us $E_d = 10$) per client. A normal FL stage in our settings trains the clients for T rounds, 1 epoch per client/round. Therefore, `stage 1` increases the time by a factor of $(\frac{E_d}{T} + 1)$. Usually, $E_d << T$, making the pre-training time overhead negligible.

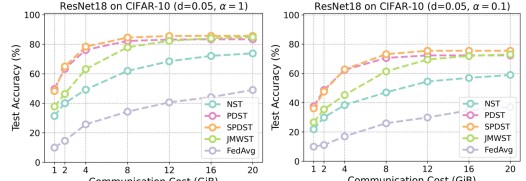

The communication overhead of `stage 1` is also negligible compared to that in each round for the `stage 2` FL training. Each participant only needs to send $L$ values for an $L$-layer

Figure 7: performance comparison at different up-link limits for (a) $\alpha = 1.0$ and (b) $\alpha = 0.1$.

model. So, $c_d$ clients will have a total communication overhead of ($L \times c_d \times 32$) bits, assuming 32-bit number representation.

**Computation saving for FLASH.** The training FLOPs for a layer $l$ ($F_{layer}^l$) can be partitioned into forward operation FLOPs ($F_{fwd}^l$), backward input ($F_{back\_in}^l$) and weight gradient ($F_{back\_wt}^l$) compute FLOPs. With the assumption of no-compute cost associated to the zero-valued weights via zero-gating logic Kundu et al. (2021a), the $F_{layer}^l$ for FLASH with parameter density $d$ ($d << 1.0$) is

$$F_{layer}^l = d \times [Fu_{fwd}^l + Fu_{back\_in}^l] + s_a \times Fu_{back\_wt}^l \tag{3}$$

where $s_a$ is $d$ and 1 for SPDST and JMWST, respectively. $Fu_x^l$ represents the corresponding FLOPs associated with an unpruned layer. Thus SPDST provides improved computation benefits along with the communication savings. Further details on FLOPs computation is provided in the Appendix.

**Performance at limited communication budget.** Fig. 7(a) and (b) show the performance of FL models when the clients are communication limited. In particular, we see both PDST and SPDST can significantly outperform other approaches in yielding a significantly well-trained model at low comm. budget. This can be attributed to their significantly smaller model sizes, helping them to run for higher number of rounds than others, on a limited bandwidth scenario.

## 6 CONCLUSIONS

This paper presented federated lottery aware sparsity hunting methodologies to yield sparse server models with low parameter density while costing insignificant accuracy drop compared to the un-pruned counterparts. In particular, we demonstrated two efficient sparse learning solutions specifically tailored for FL, enabling better computation and communication benefits over existing sparse learning alternatives. We experimentally demonstrated the superiority of our models in yielding up to $\sim 10.1\%$ improved accuracy with $\sim 10.26 \times$ fewer communication costs, compared to the existing alternatives Qiu et al. (2021), at similar hyperparameter settings. Future research direction of this work includes the theoretical understanding of our observations, and further empirical demonstrations on newer class of models including transformers.

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

# A APPENDIX

## A.1 MODEL ARCHITECTURES

Table 6 shows the model architectures used for MNIST and FEMNIST datasets. For CIFAR-10 we used ResNet18 He et al. (2016) with the first CONV layer kernel size as $3 \times 3$ instead of original $7 \times 7$.

## A.2 HETERO-FLASH ALGORITHM

Algorithm 2 details the training algorithm in hetero-FLASH. Note that and `aggrParamUpdateMask` and `subSampleServerModel` are the two functions that play key role in supoorting heterogeniety in sparsity ratios for different clients. The details of these two functions are elaborated in Algortihm 3 and 4, respectively. We plan to open-source our code upon acceptance of the paper.

Table 6: Architecture used for MNIST and FEMNIST datasets

| MNIST | FEMNIST |
|---|---|
| $\text{CONV}5 \times 5(C_o = 10)$ | $\text{CONV}5 \times 5(C_o = 32)$ |
| max_pool | max_pool |
| $\text{CONV}5 \times 5(C_o = 20)$ | $\text{CONV}5 \times 5(C_o = 64)$ |
| max_pool | max_pool |
| $\text{FC}(5120, 50)$ | $\text{FC}(3136, 2048)$ |
| $\text{FC}(50, 10)$ | $\text{FC}(2028, 62)$ |

---

**Algorithm 2:** Hetero-FLASH Training.

**Data:** Training rounds $T$, local epochs $E$, client set $[[\mathcal{C}_{N_1}], ..., [\mathcal{C}_{N_M}]]$, clients per rounds $c_r$, target density set $d_{set} = [d_1, ..., d_M]$, sensitivity warm-up epochs $E_d$, density warm up client count $c_d$, initial value of freeze masks $m_{freez} = 0$, training algorithm $A$ and aggregation type $Agr$.

1   $\mathcal{M}^{init} \leftarrow \texttt{createRandomMask}()$
2   $\Theta^{init} \leftarrow \texttt{initMaskedWeight}(\mathcal{M}^{init})$
3   $\underline{\texttt{serverExecute}}$:
4   Randomly sample $c_d$ clients $[\mathcal{C}_d] \subset [\mathcal{C}_{N_M}]$
5   **for** *each client $c \in [\mathcal{C}_d]$ in parallel* **do**
6     $\Theta_c \leftarrow \texttt{clientExecute}(\Theta^{init}, E_d, 0)$
7     $\mathcal{S}_c \leftarrow \texttt{computeSensitivity}(\Theta_c)$
8   **end**
9   $\mathcal{M}^0 \leftarrow \texttt{initMask}([\mathcal{S}_c], d_{set})$
10   $\Theta^0 \leftarrow \texttt{initMaskedWeight}(\mathcal{M}^0)$
11   $m_{freez} \leftarrow \texttt{freezeMask}(A)$
12   **for** *each round $t \leftarrow 1$ to $T$* **do**
13     Randomly sample $c_r$ clients $[\mathcal{C}_r] \subset [\mathcal{C}_N]$
14     **for** *each client $c \in [\mathcal{C}_r]$ in parallel* **do**
15      $\Theta_c^t \leftarrow \texttt{clientExecute}(\Theta^{t-1}, E, m_{freez})$
16     **end**
17     $\Theta_S^t \leftarrow \texttt{aggrParamUpdateMask}([\Theta_c^t], Agr)$
18     $\Theta^t \leftarrow \texttt{subSampleServerModel}(\Theta_S^t, [\Theta_c^t], d_{set}, m_{freez})$
19   **end**
20   $\underline{\texttt{clientExecute}}(\Theta_c, E, m_{freez})$:
21   $\Theta_{c^0} \leftarrow \Theta_c$
22   **for** *local epoch $i \leftarrow 1$ to $E$* **do**
23     $\Theta_{c^i} \leftarrow \texttt{doSparseLearning}(\Theta_{c^{i-1}}, m_{cfreez})$
24     $m_{freez} \leftarrow \texttt{checkUpdateMask}()$
25   **end**
26   **return** $\Theta_{cE}$

---

**Algorithm 3:** `aggrParamUpdateMask`

**Data:** Round $t$, aggregation type $Agr$ [`fedAvg`, `weightedFedAvg`], clients updates $[\Theta^t] = [\Theta_{c_1}, ..., \Theta_{c_r}]$, client data size $[ds_{c_1}, ..., ds_{c_r}]$

1   **if** *$Agr$ is `fedAvg`* **then**
2     $\Theta_S^t \leftarrow \frac{1}{\Sigma_{c_i=1}^{c_r} ds_{c_i}} \Sigma_{c_i=1}^{c_r} ds_{c_i} \cdot \Theta_{c_i}^t$
3   **else**
4     //For hetero-FLASH
5     $\mathcal{W}^t \leftarrow \texttt{initWeightFactor}()$
6     **for** *each update $\Theta_{c_i} \in [\Theta^t]$* **do**
7      $\mathcal{W}_{c_i}^t \leftarrow ds_{c_i} \times \texttt{retrieveMask}(\Theta_{c_i})$
8      $\mathcal{W}^t \leftarrow \mathcal{W}^t + \mathcal{W}_{c_i}^t$
9     **end**
10     //`safeDivide`(a,b): gives zero anywhere the b is queal to zero
11     $\Theta_S^t \leftarrow \Sigma_{c_i=1}^{c_r}[\texttt{safeDivide}(\mathcal{W}_{c_i}^t, \mathcal{W}^t) \cdot \Theta_{c_i}^t]$
12   **end**

---

**Algorithm 4:** subsampleServerModel

---

**Data:** Current round id $t$, client set $[\mathcal{C}_r]$, aggregated Weight $\Theta_S^t$ of model with $L$ layers, support density set $d_{set} = [d_1, ..., d_M]$ where $d_i < d_{i+1}$, model layer-wise parameter count $[k] = [k^1, ..., k^L]$.

1 **if** $size(d_{set})$ *is* 1 **then**
2    //JMWST subsampling in FLASH
3    $\mathcal{M} \leftarrow$ initMaskWithZeros()
4    $[\hat{d}^1, ..., \hat{d}^L] \leftarrow$ avgLayerWiseDensity($[\mathcal{C}_r]$)
5    $r_f \leftarrow \frac{d_1 \times K}{\sum_{l=1}^{L} \hat{d}^l . k^l}$
6    **for** *layer* $l \leftarrow 1$ **to** $L$ **do**
7       idx $\leftarrow$ getSortedWeightIndeices($\Theta_S^t, l$)
8       $n_z \leftarrow$ int($r_f \times \hat{d}^l \times k^l$) //number of non-zeros
9       $\mathcal{M}^l[\text{idx}[: n_z]] \leftarrow 1$
10    **end**
11 **else**
12    //For hetero-FLASH
13    **for** $d_i \in d_{set}$ **do**
14       $\mathcal{M}_i \leftarrow$ initMaskWithZeros()
15    **end**
16    $\mathcal{D}_s^t \leftarrow$ getCurrentDensity($\Theta_S^t$)
17    $[\hat{d}^1, ..., \hat{d}^L] \leftarrow$ getLayerWiseDensity($\Theta_S^t$)
18    **for** *layer* $l \leftarrow 1$ **to** $L$ **do**
19       idx $\leftarrow$ getSortedWeightIndeices($\Theta_S^t, l$)
20       **for** $d_i \in d_{set}$ **do**
21          $r_{f_i} \leftarrow \frac{d_i}{\mathcal{D}_s^t}$
22          $n_z \leftarrow$ int($r_{f_i} \times \hat{d}^l \times k^l$)
23          $\mathcal{M}_i^l[\text{idx}[: n_z]] \leftarrow 1$
24       **end**
25    **end**
26 **end**

---

## A.3 ADDITIONAL COMPARISONS

We now compare the performance of FLASH with that of yielded via FedSpa Huang et al. (2022), and FedDST Bibikar et al. (2021). For FedSpa, we implemented their proposed algorithm in our settings and kept all the hyperparameters same for an apple-to-apple comparison. For FLASH, we report the best of the accuracy yielded via models trained using SPDST and JMWST. As shown in Table 7, FLASH generated models can outperform that generated via FedSpa with an improved accuracy of up to 2.41%. Similar trend is observed when we compare with FedDST and as Table 8, on MNIST dataset, FLASH can have an accuracy improvement of up to 1.41%.

Table 7: Comparison of FLASH with FedSpa Huang et al. (2022) on CIFAR-10 with ResNet18.

| Data distribution | Method | Density ($d$) | Best Acc. (%) | $\delta_{Acc}$ |
|---|---|---|---|---|
| $\alpha = 1000$ | FedSpa | 0.05 | 85.63 | – |
| | FLASH | 0.05 | **87.18** | +1.55 |
| $\alpha = 0.1$ | FedSpa | 0.05 | 73.08 | – |
| | FLASH | 0.05 | **75.49** | +2.41 |

## A.4 MORE QUANTITATIVE ANALYSIS

**1. Ablation with the mask update interval rounds ($r_{int}$).** As mentioned in the original manuscript, in the case of JMWST, to save communication energy we often can choose not to update the mask every round. We thus performed ablation with an increased frequency of mask update interval round $r_{int}$ from the default value of 1 (similar to Qiu et al. (2021)). Table 9 shows the results with different $r_{int}$. In particular, as we can see in the table, less frequent update intervals does not degrade the final

Table 8: Comparison of FLASH with FedDST Bibikar et al. (2021) on pathologically non-IID MNIST. For this comparison we used same hyperparameter settings and models as that in Bibikar et al. (2021).

| Method | Density ($d$) | Communication Cost (GiB) | Best Acc. (%) | $\delta_{Acc}$ |
|---|---|---|---|---|
| FedDST | 0.2 | 1.0 | 96.10 | – |
| FLASH | | | **97.51** | +1.41 |
| FedDST | 0.2 | 2.0 | 97.35 | – |
| FLASH | | | **97.69** | +0.34 |

model performance. Moreover, a less frequent update can provide additional savings in terms of up-link cost for the models as the masks do not change every round.

Table 9: Ablation with different mask update intervals for JMWST for a target density $d = 0.1$ on CIFAR-10.

| Model | Data distribution | Mask update interval rounds ($r_{int}$) | | | |
|---|---|---|---|---|---|
| | | $r_{int} = 1$ | $r_{int} = 2$ | $r_{int} = 5$ | $r_{int} = 10$ |
| | IID ($\alpha = 1000$) | $87.62 \pm 0.35$ | $87.76 \pm 0.07$ | $87.86 \pm 0.13$ | $87.67 \pm 0.09$ |
| ResNet18 | non-IID ($\alpha = 1$) | $86.45 \pm 0.31$ | $86.26 \pm 0.07$ | $86.36 \pm 0.13$ | $86.68 \pm 0.25$ |
| | non-IID ($\alpha = 0.1$) | $74.74 \pm 1.07$ | $73.73 \pm 1.18$ | $75.47 \pm 1.08$ | $77.14 \pm 0.22$ |

**2. Impact of Number of participating clients per round.** Fig. 8 (a), shows that JMWST and SPDST follow the same pattern at the baseline model ($d = 1.0$) with FedAvg. In other words, similar to FedAvg, as the $c_r$ increases, the performance enhances. Also, for a specific $c_r$, JMWST and SPDST perform better than PDST and NST.

**3. Impact of Batch-Normalization layer statistics.** Fig 8 (b), shows the performance comparison between batch bormalization (BN) and static batch normalization (static BN, as suggested in Diao et al. (2020)). In particular in our setting, using BN layer statistics always outperform the static BN.

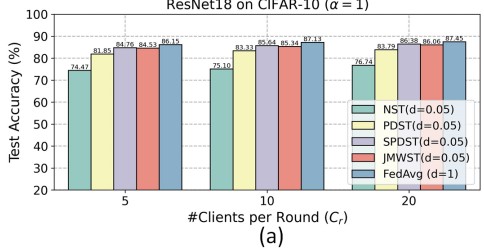 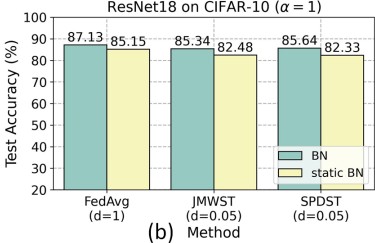

Figure 8: (a) Performance of the final trained model for different participating clients per round, (b) Significance of BN and Static BN in final model performance.

**4a. Revisiting sparse mask mismatch for NST with VGG16.** Fig. 9 shows the comparison of SM between centralized and FL settings with NST on VGG16, another popular model variant. Similar to our observed trend with ResNet18, we see a significantly high SM for FL settings with a target $d - 0.05$. This strengthens the generality of our observed limitations across different class of DNN models.

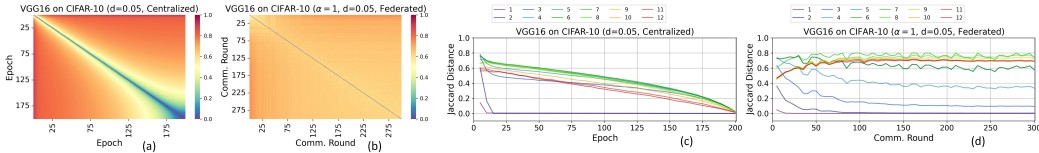

Figure 9: (a)-(b) Sparse mask mismatch (SM) for VGG16 in (a) centralized and (b) FL settings with NST. (c)-(d) Layer-wise SM vs. training epochs (rounds) for VGG16 in (c) centralized and (d) FL settings, respectively, with NST.

**4b. Revisiting sparse mask mismatch for FLASH.** As demonstrated in Figs 10, the sparse mask mismatch in the case of JMWST significantly reduces, helping the mask train in a convergent way, significantly faster than that in NST.

Fig. 11 shows the layer-wise SM, for centralized trained model (Fig. 11a) and FL trained model with sparsity (Fig. 11b-c). In particular, the SM at later layer can significantly reduce in the case of JMWST as compared to NST, further demonstrating the convergence ability even at the later layers.

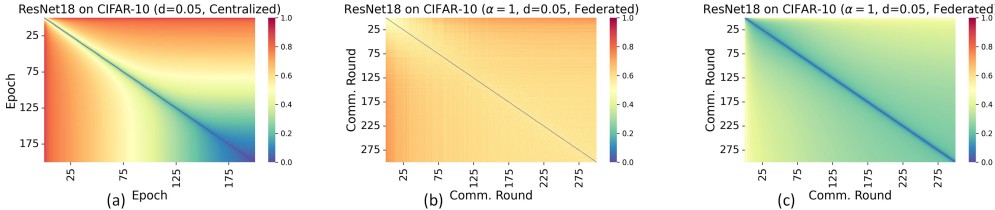

Figure 10: Sparse mask mismatch (SM) for (a) centralized sparse learning, (b) NST, and (c) JMWST in federated settings.

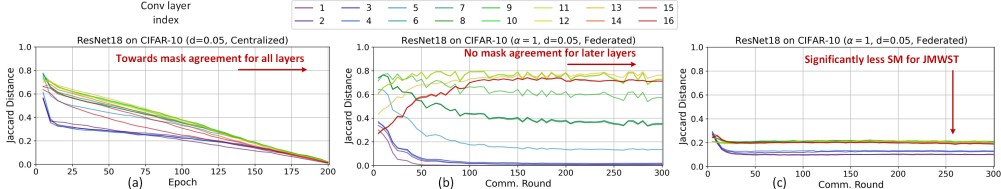

Figure 11: Layer-wise sparse mask mismatch (SM) vs. training epochs (rounds) plot for (a) centralized and (b) FL with NST, and (c) FL with JMWST.

**4c. Sparse mask mismatch as a function of $d$.** To understand the relation of SM with $d$, we performed the baseline sparse training (NST) with ResNet18 on CIFAR-10 for three different target densities, 0.05, 0.25, 0.5. As shown in Fig. 12, the SM tends to reduce for higher density. In particular, Fig. 12(d) shows the SM for CONV layer 16 (a later layer), after round 200. The SM reduces by $1.53\times$ for $d = 0.5$ than that with $d = 0.05$, strengthening our general observation that SM becomes prominent as the density gets lower.

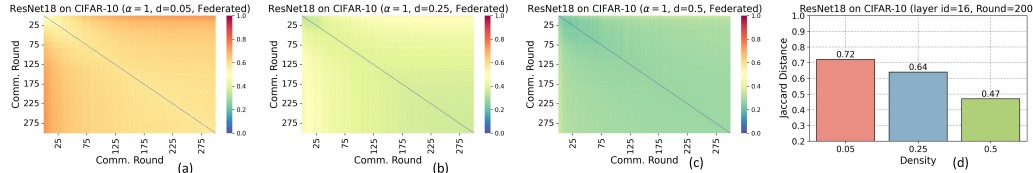

Figure 12: (a-c) SM for FL settings for three different $d$ of 0.05, 0.25, and 0.5, respectively. (d) Comparison of Jaccard distance values for the $16^{th}$ CONV layer of ResNet18 after round 200 for different $ds$.

**4d. Sparse mask mismatch as a function of number of clients.** To understand the relation of SM with number of total clients, we performed the baseline sparse training (NST) with ResNet18 on CIFAR-10 for 50 and 200 clients, respectively. As shown in Fig. 13, the SM concern persists, irrespective of the number of clients. This strengthens the generality of our observations over total number of clients.

**5. Convergence trend of proposed algorithms.** Fig. 14 shows the test accuracy vs FL rounds for NST, PDST, SPDST, and JMWST algorithms on CIFAR-10 dataset with non-IID data distribution ($\alpha = 1$). As shown in the plots, for $d = 0.05$ and $d = 0.1$, NST has slower convergence with lower final accuracy. Introducing consensus among the clients for the sparse mask accelerates the convergence and enhances the final performance.

**6. Communication saving and FLOPs evaluations.** Employing sparse learning in FL helps participating clients reduce both the communication and compute costs (FLOPs) for training. Without

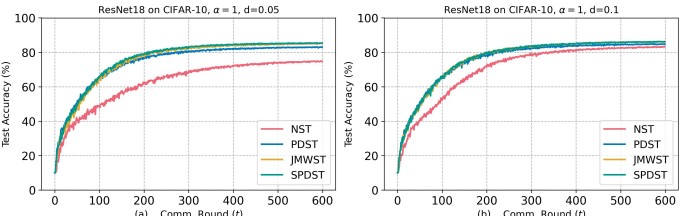

Figure 13: (a-b) SM for FL settings for (a) 50 and (b) 200 clients. (c-d) Layer-wise SM vs. training rounds for (c) 50 and (d) 200 clients.

Figure 14: Performance of proposed algorithms vs. comm. rounds on CIFAR-10 dataset for (a) $d = 0.05$ (b) $d = 0.1$.

the loss of generality, we now evaluate the convolutional layer training FLOPs for FLASH, and demonstrate the relation of parameter density $d$ with FLOPs and communication saving. Let us assume a layer $l$ has a weight tensor $\boldsymbol{\theta}^l \in \mathbb{R}^{C_o \times C_i \times h \times w}$, where $h$ and $w$ are the height and width of the convolutional kernel, and $C_o$ and $C_i$ represent the number of filters and channels per filter, respectively. It takes an input tensor $\boldsymbol{I} \in \mathbb{R}^{C_i \times H \times W}$ to produce an output tensor $\boldsymbol{O} \in \mathbb{R}^{C_o \times R \times S}$. Let us also assume $d$ to be the density of non-zero in the weight tensor for all the layers. During training, FLOPs associated to each weight tensor update can be partitioned in to three component, namely, forward pass FLOPs ($F_{fwd}$), backward input grad FLOPs ($F_{back\_in}$), and backward weight grad FLOPs ($F_{back\_wt}$). The weight sparsity helps both $F_{fwd}$ and $F_{back\_in}$ to reduce proportionally as given below.

$$F_{fwd} = d \times C_i \times (h \times w) \times (R \times S) \times C_o \tag{4}$$

$$F_{back\_in} = d \times C_i \times (h \times w) \times (H \times W) \times C_o \tag{5}$$

Finally, if the zero weights' gradients flow is computed for the purpose of mask learning, then $F_{back\_wt}$ can't leverage the advantage of low parameter density. Thus during mask training of JMWST, as the gradient needs to be dense, $F_{back\_wt}$ is same as that in dense computation. However, for SPDST, zero weights remain as zero, allowing us to safely skip the associated gradient computation. This essentially helps SPDST to extract benefits of sparsity during all the three stages of FLOPs computation. Following Eq. shows the $F_{back\_wt}$ in FLASH.

$$F_{back\_wt} = s_a \times C_i \times (h \times w) \times (H \times W) \times C_o \tag{6}$$

where,

$$s_a = \begin{cases} 1, & \text{for JMWST} \\ d, & \text{for SPDST} \end{cases} \tag{7}$$

For similar density, clients in ZeroFL also enjoys similar benefits in $F_{fwd}$ and $F_{back\_in}$. However, $F_{back\_wt}$ can be reduced only via introduction of activation sparsity, $a$. It is well surveyed in literature that having sparse activation density as low as parameter density may significantly impact model performance. Thus generally, $a > d$ that allows SPDST to enjoy a FLOPs benefit of $\frac{a}{d}$ for $F_{back\_wt}$.

These computations can be easily extended to linear layers. Also, we can safely ignore the FLOPs associated to the BN layers due to their negligible contribution to the total FLOPs.

**7. Discussion on compute benefits at the edge.** To extract FLOPs benefits for irregular pruning in FLASH, we assume that the com-

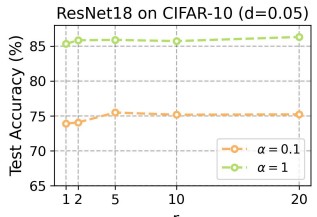

Figure 15: Test accuracy vs. mask update interval round.

pute energy for the sparse network can be avoided via the means of clock-gating Yang & Kim (2018) of the zero-valued weights. Moreover, there has been recent development of sparsity-friendly DNN

Figure 16: Computation and Communication relation with (a) each other (b, c) with different density levels for SPDST algorithm.

accelerators Qin et al. (2020) that can efficiently reduce the compute cost by a significant margin. Such accelerators can leverage the yielded sparse FL models to deploy at compute constrained edges.

**8. FLOPs vs. communication cost for different density budget.** To reach a target accuracy value, we now plot the FLOPs to uplink communication cost for different density budget in Fig. 16.

**9. Impact of $r_{int}$.** As depicted in Fig. 15, the test accuracy improves with the increase in JMWST mask update interval $r_{int}$. In particular, for both $\alpha = 0.1$ and $1.0$, the accuracy with increased $r_{int}$ can be up to ~1.6% and ~0.98%, respectively. Interestingly, the improvement tend to saturate after certain $r_{int}$. Thus, we consider important sparse learning hyperparameter search as an interesting future research direction.

## A.5 DISCUSSION ON SUPPORT FOR HARDWARE-FRIENDLY SPARSITY PATTERNS

Irregular sparsity often are not well suited for hardware benefits without any dedicated architecture or compiler support. Among the various hardware-friendly sparsity patterns recently proposed $N : M$ sparsity Zhou et al. (2021) has gained significant attention, due its less stricter constraints compared to other structured sparsity patterns. For SPDST, post stage 1 sparse mask selection can be easily extended to support the $N : M$ sparsity. In particular, for a layer $l$, instead of random assignment of $d^l \times k^l$ non-zero mask locations, we can partition the total non-zero elements in to $G^l$ groups, where each group will contain $d^l \times k^l/G^l$ non-zero elements. Here $G^l$ is evaluated as $k^l/M$, $M$ representing the total element size out of which we need to have a certain fraction as non-zero, and $k^l$ represents the total number of weights for that layer. As the masks remain frozen, we are ensured such pattern is maintained throughout the training for each client to extract the benefit. For JMWST, we can adapt this principle in the prune and regrow policy that happens during local training of each client. We have added a section in the appendix detailing on this important discussion.

