# OpenReview forum: "Lottery Aware Sparsity Hunting: Enabling Federated Learning on Resource-Limited Edge"
_ICLR.cc/2023/Conference — Submitted to ICLR 2023_

### Official Review · Reviewer_cFmd · 2022-10-23

**Confidence:** 3
**Correctness:** 2
**Technical Novelty And Significance:** 2
**Empirical Novelty And Significance:** 2
**Recommendation:** 3

**Clarity, Quality, Novelty And Reproducibility:**

I think the motivation and idea would be better explained if the authors could provide a rigorous description of the baseline sparse federated learning algorithm in Section3.

**Strength And Weaknesses:**

## strength
The authors identify an interesting problem: the accuracy drops for sparse training in the federated learning setting.

## weakness
1. The paper is hard to follow. I don't understand why FL cannot achieve converged sparse masks. The only difference between centralized and federated learning is data distribution, but you can always have a consensus model on the server by averaging the gradients from workers, right? Or if you use local SGD and average the models, you can fix the sparse mask on each worker between synchronizations, right?

2. The proposed algorithm is designed based on one particular pruning method (Kundu et al. 2021). Does it apply to other sparse training methods (e.g. RigL)?

**Summary Of The Paper:**

This paper studies sparse training in a federated learning setting. The authors show that a naïve implementation has significant accuracy degradation, and they proposed a method called federated lottery-aware sparsity hunting. Experiments on ResNet-18 on MNIST, EMNIST, and CIFAR-10 show that the proposed method improves accuracy and reduces communication costs.

**Summary Of The Review:**

The setting of this work is unclear to me. The applicability of the proposed method also seems narrow.

---

### Official Review · Reviewer_vopV · 2022-10-24

**Confidence:** 3
**Correctness:** 3
**Technical Novelty And Significance:** 2
**Empirical Novelty And Significance:** 2
**Recommendation:** 6

**Clarity, Quality, Novelty And Reproducibility:**

(1) Is the model performance sensitive to the number of clients? How the model accuracy and SM changes as the number of clients vary?

(2) Can you also show SM mismatch (similar to Figure 3) for modest density ratio? If I understand correctly you are claiming that such problem only exists for low density ratio?

(3) Similarly, is there any breaking point for the density ratio in which the SM mismatch ratio emerges? What I am trying to understand is to better understand the core reason behind this SM mismatch ratio. Is this just a byproduct of higher probability of mismatch as the number of zeros increase?

(4) (not looking for results) Do you have any intuition how your method can be applied to other forms of sparsity? For example, as you may be familiar, structure sparsity (e.g. N:M, Pixelated Butterfly) methods are more hardware-friendly and efficient for both inference and training. Do you have any intuition how this SM mismatch affects these forms of sparsity?

**Strength And Weaknesses:**

**Strength**

- The paper starts off with multiple observations about the meager accuracy of models in a sparse federated learning setting and then proposes multiple solutions to mitigate the challenges.

- The results across few models seem to be promising, both in terms of final model accuracy and reduced communication.

**Weaknesses**

- While I liked the structure of the paper, it is not clear how bringing the heterogeneity of a system is aligned with the message of the paper.

- While sparsity is an interesting paradigm to reduce various bottlenecks in models, I am not sure if that low density rate (where the most benefit of the paper emerges) at the cost of accuracy degradation would be an acceptable solution even for federated learning. As such, this is not clear if the proposed solution is for a realistic problem setting.

**Summary Of The Paper:**

The paper targets sparse training in the federated learning domain. The paper starts with multiple observations (on small models) about how disagreement of clients on the sparsity mask pattern could significantly reduce the model accuracy, compared to a centralized training. It also shows that this disagreement on the sparsity mask pattern diverges as the training progresses. Based on these observations, the authors propose a two-stage sparse federated learning method with two main purposes; (1) decoupling the identification of initial sparse mask from training and (2) collaboratively learning sparse mask and model weights. Finally, the paper extends the method for a heterogeneous environment (which I think may not be aligned with the main message of the paper) where each client has a different resource budget for which different sparsity mask is required.

**Summary Of The Review:**

I think the paper is well-motivated with clear description of the observations. The authors went deep into a better understanding of the root cause of low model accuracy in a federated learning setting. While I agree with the authors that this disagreement between sparsity mask patterns *could* be one of the reasons for low accuracy, but I am not convinced that this is the sole reason for such performance degradation. In addition, as mentioned in the questions, it would be great if the authors could provide additional results for higher density ratio to better understand whether such problem exists in a more realistic setting.

---

### Official Review · Reviewer_womR · 2022-10-31

**Confidence:** 3
**Correctness:** 3
**Technical Novelty And Significance:** 3
**Empirical Novelty And Significance:** 3
**Recommendation:** 5

**Clarity, Quality, Novelty And Reproducibility:**

*Clarity*: The methodology in the paper is somewhat hard to follow due to 1) the missing preliminary knowledge for specifying the Federated learning setting, 2) the lack of an overall framework or formal formulation of the proposed method, and 3) the mixup between algorithm lines and descriptions.

*Novelty*: The proposed method seems novel to me owing to its two-stage sparse mask training method and the practical heterogeneous device budget setting. However, it remains unclear to me what is the key contribution of this work to address the observed limitations. The main technical concern is whether the proposed method can be used in very deep neural networks (like over 100 layers and the recent transformer-based architecture).

**Strength And Weaknesses:**

Pros:
- The proposed hetero-FLASH method is interesting and well adapted to the Federated learning setting over diverse edge resources. While the sampling and fusion strategy is straightforward, the proposed solution poses a good baseline for exploring dynamic budgets in a Federated learning framework.
- It is technically sound to bridge the gap between sparse network training and its dense counterpart by developing a two-stage method to evaluate mask sensitivity in advance.
- Two mask learning strategies, namely SPDST and JMWST, were designed and developed on three datasets under two different settings.

Cons:
- While several observations were discussed and provided with empirical evidence, it somewhat lacks in-depth or theoretical analysis regarding the reasons behind these observations. Also, all the observations were provided on the same model architecture (ResNet18), which might lead to model bias in the conclusions.
- The proposed stage 1 lies in the key contribution of this work. Thus, some ablated models like SPDST w/o state 1, and JMWST w/o state 1 are expected to be provided in the experiment. Plus, it is unclear to me how the proposed method encourages mask convergence -- by individually using SPDST or JMWST, or by dynamically switching between these two methods?
- The experiment results are less convincing due to the lack of baselines (e.g., Huang et al. (2022) and Bibikar et al. (2021)) and ablated models. Although mask convergence has been well discussed, training convergence and parameter analysis on mask initialization are expected. It also remains unclear if the proposed method can be applied in a deeper model, such as ResNet34, ResNet50, and ResNet101

**Summary Of The Paper:**

The paper tries to improve the sparse network training efficiency in a Federated learning framework in two folds: 1) bridging the gap between the sparse network and its dense counterpart (e.g., FedAvg) and 2) saving the communication cost between clients and the server.   To this end, the proposed method performs a two-stage sparse training, including the mask-sensitive evaluation (for mask initialization) and networking training, and investigates two different mask learning strategies -- 1) fixed masks and 2) jointly training over masks and weights. Experiments on three public datasets were provided in terms of both IID and non-IID settings.

**Summary Of The Review:**

Overall, the paper provides a technically sound method to solve several practical challenges in incorporating sparse network training into a Federated learning framework. However, please refer to my concerns on methodology, experiment, and network depth in the above sections.

---

### Decision · Program_Chairs · 2023-01-20

**Decision:**

Reject

**Justification For Why Not Higher Score:**

see above.

**Justification For Why Not Lower Score:**

see above.

**Metareview: Summary, Strengths And Weaknesses:**

There was an extensive discussion with the reviewers about this paper but all reviewers were convinced that the paper was not ready for publication in its current form. The core issues were summarized by a reviewer in a Dec 7th, 2022 entry, but we give a very brief recap of the issues here.

1) mismatch of the heterogeneous with your work.

2) hetero-FLASH does not align with the core message of the paper.

3) Low coverage of the density set.

4) narrow applicability of approach while insisting otherwise.

5) Lack of applicability of efficient extreme sparsity in modern hardware capabilities.

6) issues with accuracy.

7) baseline comparisons should be improved. Need stronger baselines.

8) Questions about the number of clients, in particular accuracy as a function of clients needs to be better explored.


The ICLR SAC would also like to make a recommendation to the authors for the future, and that is the lesson that sometimes "less is indeed more." The abundance of text and endless series of new results that you offered in response to the reviewers did not do you any favors. It would have been better to be more succinct and to the point. A paper needs to be judged largely by the initial submission, and there is a page length in the original submission for a reason.  Openreview with its essentially unlimited space gives people the impression that they can write an essentially unlimited length rebuttal. But this is not always beneficial, and sometimes it is the opposite. If you are convinced in your papers merit, please boil down the essential results from this lengthy rebuttal and place them carefully in the next finite-length version of your paper and re-submit to the next conference.

**Summary Of Ac-Reviewer Meeting:**

see above.